# Examination of Inner Retinal Layers in Unilateral Wet Age-Related Macular Degeneration Treated with Anti-VEGF, Compared to Fellow Untreated Eyes

**DOI:** 10.3390/ijms24010402

**Published:** 2022-12-26

**Authors:** Małgorzata Wichrowska, Sławomir Liberski, Anna Rzeszotarska, Przemysław Wichrowski, Jarosław Kocięcki

**Affiliations:** 1Department of Ophthalmology, Poznan University of Medical Sciences, ul. Augustyna Szamarzewskiego 84, 61-848 Poznan, Poland; 2Doctoral School, Poznan University of Medical Sciences, ul. Bukowska 70, 60-812 Poznan, Poland; 3Clinic of Anesthesiology and Intensive Therapy, Poznan University of Medical Sciences, ul. Długa 1/2, 61-848 Poznan, Poland

**Keywords:** age-related macular degeneration, ganglion cell layer, retinal nerve fiber layer, anti-VEGF

## Abstract

The main aim of this study was to characterize the retinal nerve fiber layer (RNFL) and ganglion cell layer (GCL) thickness in the macular area eyes affected by wet age-related macular degeneration (wAMD) treated with anti-VEGF and compare the results with the control of fellow untreated eyes affected by early stages of dry age-related macular degeneration (dAMD). Additionally, we aimed to estimate if the number of injections received and other factors, including age, best-corrected visual acuity (BCVA), or sex, may affect the differences in the obtained measurements of retinal nerve fiber layer thickness. We prospectively included 106 eyes of 53 patients with unilateral wet age-related macular degeneration. The fellow eyes with non-advanced dry age-related macular degeneration served as a control group in a cross-sectional study. RNFL and GCL in the macular region were evaluated using optical coherence tomography, with outcomes expressed as differences in the thickness of both examined layers between the study and control groups. We found thinner GCL in wAMD vs. dAMD (*p* < 0.001). In turn, the RNFL layer did not show any statistically significant differences between the two groups (*p* = 0.409). Similarly, we found a statistically significant correlation between the number of injections and the layer thickness (*p* = 0.106). Among all assessed parameters, age over 73 was the only factor significantly affecting the thickness of the retinal nerve fiber layer in both groups (*p* = 0.042). The morphology of the inner layers of the retina in dry and wet AMD seems to differ, possibly due to differences in the etiopathogenesis of these two forms of the disease. In our study, the retinal ganglion cell layer was thinner in the treated vs. fellow eye (with dry AMD), while the nerve fiber layer was not significantly different between the groups. The number of anti-VEGF injections had no effect on the thickness of the macular nerve fiber layer.

## 1. Introduction

Age-related macular degeneration (AMD) is thought to affect up to a quarter of the population over the age of 75 [1]. In the population over 60, it is considered the third most common cause of blindness (after cataracts and glaucoma) [2]. AMD is a disease of the retina that affects the macula, resulting in impairment of central vision. The two currently recognized forms of the disease are dry AMD (dAMD), characterized mainly by the presence of drusen and disturbances within the retinal pigment epithelial layer, and wet AMD (wAMD), the exudative form exhibiting macular neovascularization [3,4]. 

The etiopathology of age-related macular degeneration is still not fully understood [3,5]. It is suggested that the disease process begins mainly at the level of the outer retinal layer and is associated with the senescence of retinal pigment epithelial cells. This results in their impaired metabolism, Bruch’s membrane thickening, and defective choroidal perfusion [3,4,6,7,8,9]. Cell aging can trigger the secretion of biologically active factors such as interleukins IL-1α, IL-1β, IL-6, and IL-8, monocyte chemotactic proteins MCP-1 and MCP-2, granulocyte and macrophage granulocyte colony growth factors GCSF and GMCSF, which can stimulate the tissue complement system, microglial cells, and macrophages. This, in turn, can result in tissue remodeling and disruption of local homeostasis [10]. In addition, one of the dominant components of oxidized lipoproteins in drusen, present within the Bruch’s membrane in wAMD, seems to be 7-ketocholesterol, which can activate microglia by releasing cytokines, indirectly resulting in increased expression of vascular endothelial growth factor (VEGF) in RPE cells. Secreted VEGF promotes endothelial cell migration, resulting in MNV formation [10].

Another hypothesis regarding the etiopathology of AMD focuses on microvascular abnormalities within retinal vascular plexuses, with potential anterograde transsynaptic degeneration [10,11,12]. The chronology of events in the development of the disease remains unclear. Some authors indicate that it can also result in damage to further neurons of the visual pathway, including the inner layers of the retina: the retinal nerve fiber layer (RNFL) and the retinal ganglion cell layer (GCL). However, the data regarding this pathomechanism are still sparse [4,13,14,15,16].

Although there is still no generally available cure for the dry form of the disease, the wet form has been successfully treated with anti-VEGF intravitreal injections for many years [10]. In Polish ophthalmology centers, the most commonly used anti-VEGF agents include aflibercept, ranibizumab, and bevacizumab, and recently brolucizumab. Ranibizumab, a humanized monoclonal antibody fragment, is capable of binding all VEGF isoforms. Aflibercept, a fusion protein composed of the human IgG Fc fragment and the second and third domains of human VEGF receptors (VEGFR-1 and VEGFR-2), acts as an inhibitor of both VEGF-A and VEGF-B, and placental growth factors 1 and 2 (PIGF-1 and PIGF-2). Furthermore, bevacizumab is a full-length, humanized VEGF monoclonal antibody, while brolucizumab is a humanized antibody fragment, and like ranibizumab and bevacizumab, inhibits only VEGF-A. In recent months, faricimab, the first bispecific monoclonal antibody that can block both the activity of VEGF and angiopoietin-2 (Ang-2), has also appeared on the Polish market [2,10].

Anti-VEGF administration is recognized as an effective therapeutic strategy in wAMD. However, vascular endothelial growth factor (VEGF), high levels of which (among other factors, i.e., platelet-derived growth factor (PDGF)) are suspected to trigger macular neovascularization, is considered to play other crucial roles in the mammalian organism. It has been proven that VEGF exhibits a range of potentially positive effects, including neuroprotective and anti-apoptotic effects [3,17,18]. Therefore, at least theoretically, intravitreal anti-VEGF therapy may negatively affect retinal neurons, especially those in layers of the retina in direct contact with the administered drug.

While most authors focus on the assessment of the nerve fiber layer within the optic nerve head (the peripapillary retinal nerve fiber layer—pRNFL), the nerves contained in this region originate from the entire retina [19,20,21]. We assume that knowledge of macular change morphology may contribute to a better understanding of the nature of the disease.

Furthermore, it could be a factor of differential diagnosis with, e.g., glaucoma, which is also more prevalent in this age group [22].

The aim of this study was to assess the differences in RNFL and GCL thickness within the macula region between both disease subgroups. Our second aim was to estimate factors that may affect the RNFL thickness, including visual acuity, sex, age, the number of anti-VEGF injections administered, and whether and how changes in the retina nerve fiber layer thickness in the macular region affect the thickness of the retinal ganglion cell layer.

## 2. Results

### 2.1. Study of the Treatment Effect on Retinal Nerve Fiber Layer Thickness

The aim of the examination was to examine the significance of differences in the retinal nerve fiber layer (RNFL) thickness in the macular area between the test and control groups, including the significance of the RNFL thickness change effect based on the number of injections. 

The model was specified either using formulations at both levels:

Level one: $Y_{ij}=a_{i}+\beta_{0}\cdot group+\beta_{1}\cdot anti-VEGF+\epsilon_{ij},\;\mathrm{where}\;\epsilon_{ij}\sim \;N\left(0,\;\sigma^{2}\right)$

Level two: $a_{i}=\alpha_{0}+u_{i},\;\mathrm{where}\;u_{i}\;\sim N\left(0,\;\sigma_{u}^{2}\right)$

Or as a composite model: 

$Y_{ij}=\alpha_{0}+\beta_{0}\cdot group+\beta_{1}\cdot anti-VEGF+u_{i}+\epsilon_{ij}$ (model 1)

In this model, retinal nerve fiber layer thickness $,\;Y_{ij}$ (x ∈ Z^+^), was a function of two covariates, the dichotomous group variable {treatment, control} and the continuous anti-VEGF variable (x ∈ Z^+^). The *a_i_* is the true mean response of observations for subject *i*. In turn, α_0_ is the grand mean—the true mean of all observations across the entire sample, $\sigma^{2}$ is the intra-subject variability, $\sigma_{u}^{2}$ is the inter-subject variability, and $\beta_{0},\;\beta_{1}$ are the slopes for the appropriate covariates.

The nominal variable was specified by dummy coding. Using the composite model specification, the model was fit with the results shown in Table 1.

The model’s total explanatory power was substantial (*R^2^_conditional_* = 0.42), and the part related to the fixed effects alone (*R^2^_marginal_*) was 0.02. The model’s intercept, corresponding to the treatment group and zero injections, was at 41.01 μm (*95% CI* (37.89, 44.14), *t*(101) = 26.07, *p* < 0.001).

The control group had a 1.35 μm thinner nerve fiber thickness in the macular area compared with the treatment group; in addition, each injection performed reduced the fiber thickness by 0.17 μm, but all these effects were not significant.

Predicted values of the RNFL thickness based on the group and anti-VEGF terms are shown in Figure 1.

### 2.2. Examination of the Effects of Retinal Nerve Fiber Layer Thickness Change on the Thickness of the Retinal Ganglion Cell Layer in Untreated/Treated Groups

The purpose of the analysis was to investigate the effect and significance of the change in RNFL thickness on the retinal ganglion cell layer (GCL) thickness score for both the study and the control groups.

The model was specified either using formulations at both levels:

Level one: $Y_{ij}=a_{i}+\beta_{0}\cdot group+\beta_{1}\cdot RNFL\;thickness+\epsilon_{ij},\;\mathrm{where}\;\epsilon_{ij}\sim \;N\left(0,\;\sigma^{2}\right)$

Level two: ai=α0+ui, where ui ~N0, σu2

Or as a composite model:

$Y_{ij}=\alpha_{0}+\beta_{0}\cdot group+\beta_{1}\cdot RNFL\;thickness+u_{i}+\epsilon_{ij}$ (model 2)

In this model, retinal ganglion cell layer thickness (GCL)$,\;Y_{ij}$ (x ∈ Z^+^), was a function of two covariates, the dichotomous group variable {treatment, control} and the continuous RNFL thickness variable (x ∈ Z^+^). The *a_i_* is the true mean response of observations for subject *i*. On the other hand, α_0_ is the grand mean—the true mean of all observations across the entire sample, $\sigma^{2}$ is the within-subject variability, $\sigma_{u}^{2}$ is the between-subject variability, and $\beta_{0},\;\beta_{1}$ are the slopes for the appropriate covariates. 

The nominal variable was specified by dummy coding. Using the composite model specification, the model was fit with the results shown in Table 2.

The model’s total explanatory power was substantial (*R^2^_conditional_* = 0.70), and the part related to the fixed effects alone (*R^2^_marginal_*) was 0.26. The model’s intercept, corresponding to treatment group and zero RNFL thickness, was at 41.44 μm (*95% CI* (39.24, 46.64), *t*(101) = 16.95, *p* < 0.001).

The control group had a 2.06 μm thicker retinal ganglion cell layer compared with the treatment group. An increase in RNFL thickness by 1.0 μm resulted in an increase in retinal ganglion cell layer thickness by 0.4 μm

Predicted values of retinal ganglion cell layer thickness based on the group and RNFL thickness terms are shown in Figure 2.

The data in Figure 2 show a linear positive relationship between RNFL thickness and ganglion cell layer thickness. The difference between groups in retinal ganglion cell layer thickness (μm) was more than 2 μm. For the 20 μm RNFL, the predicted thickness of the ganglion cell layer was 52.35 μm (*CI 95%* (49.56, 55.16)) for the treatment group and 54.42 μm (*CI 95%* (51.55, 57.29)) for the control group.

When RNFL increased to 50 μm, the predictions of thickness of the ganglion cell layer increased to 64.24 μm (*CI 95%* (62.25, 66.23)) for the treatment group and 66.30 μ (*CI 95%* (64.37, 68.24)) for the control group, respectively.

### 2.3. Examination of the Influence of BCVA, Age, and Sex on Retinal Nerve Fiber Layer Thickness

The aim of this examination was to estimate the effects of BCVA factors, sex, and age on the retinal nerve fiber layer thickness. The age factor was considered in the form of a dichotomous variable, with a breakdown relative to the mean value.

The model was specified either using formulations at both levels:

Level one: $Y_{ij}=a_{i}+\beta_{0}\cdot BCVA+\beta_{1}\cdot sex+\beta_{2}\cdot age+\epsilon_{ij},\;\mathrm{where}\;\epsilon_{ij}\sim \;N\left(0,\;\sigma^{2}\right)$

Level two: ai=α0+ui, where ui ~N0, σu2

Or as a composite model:

$Y_{ij}=\alpha_{0}+\beta_{0}\cdot BCVA+\beta_{1}\cdot sex+\beta_{2}\cdot age+u_{i}+\epsilon_{ij}$ (model 3)

In this model, retinal nerve fiber layer thickness (RNFL thickness)$,\;Y_{ij}$ (x ∈ Z^+^), was a function of three covariates: the dichotomous age {≤73 years, >73 years} and sex {female, male} variables, and the continuous BCVA variable (x ∈ ℝ^+^). The *a_i_* is the true mean response of observations for subject *i*. In turn, α_0_ is the grand mean—the true mean of all observations across the entire sample, $\sigma^{2}$ is the intra-subject variability, $\sigma_{u}^{2}$ is the inter-subject variability, and $\beta_{0},\;\beta_{1}$ are the slopes for the appropriate covariates.

The nominal variables were specified by dummy coding. Using the composite model specification, the model was fit with the results shown in Table 3.

The model’s total explanatory power was substantial (*R^2^_conditional_* = 0.39), and the part related to the fixed effects alone (*R^2^_cmarginal_*) was 0.09. The model’s intercept, corresponding to women of ≤73 years of age and zero BCVA, was at 43.81 μm (*95% CI* (39.05, 48.58), *t*(100) = 18.23, *p* < 0.001).

Results in Table 3 show that a one-unit increase in BCVA resulted in a 2.74 μm decrease in RNFL thickness, with men exhibiting a 2.62 μm thinner RNFL compared with women. However, all these differences were not significant at the 95% confidence level.

The only significant difference in RNFL thickness was shown between age groups. The group of subjects older than 73 years had a 3.35 μm thinner RNFL than the group of subjects 73 years or younger.

Predicted values of RNFL thickness based on BCVA, age, and sex are shown in Figure 3. 

Predictions for RNFL thickness based on the BCVA lower and upper sample size range {0.2, 1.0} within age and sex variables are shown in Table 4. It shows that the difference between age groups in RNFL thickness was more than 3 μm.

## 3. Discussion

In this study, we tried to characterize the differences within the inner layer of the retina (the retinal nerve layer thickness (RNFL) and the ganglion cell layer thickness (GCL) between eyes with wet age-related macular degeneration (wAMD) treated with anti-VEGFs, and the fellow eyes with dry age-related macular degeneration (dAMD). We found no difference in macular retinal nerve fiber thickness between both groups (*p* = 0.409), while GCL was thinner in the wAMD group (*p* < 0.001).

Data on this topic in the available literature vary. Most researchers found a lack of significant differences between retinal nerve fiber layer thickness of wAMD eyes treated with anti-VEGFs vs. fellow untreated eyes [19,23,24,25]. On the contrary, Zucchiatti et al. found a significantly thinner RNFL in wAMD than dAMD (interestingly, even with macular atrophy) [26]. However, all these authors analyzed the thickness of the peripapillary retinal nerve fibers (pRNFL), as this approach is much more common that that taken by other studies based on macular scans [14]. In accordance with our findings, Ilkay et al., in their studies of the macular region, found no difference in RNFL thickness between the wAMD and dAMD groups (and healthy controls) [14]. In turn, Lee et al., in an analysis of OCT scans centered on the fovea, also revealed a lack of statistical significance regarding RNFL values between eyes with different types of unilateral wAMD (with CNV and PCV) and fellow eyes [27]. These results may confirm the hypothesis that it is mainly the outer layers of the retina that are damaged by age-related macular degeneration. 

The result of our study did not meet our expectations, as we assumed that RNFL would be thinner in wAMD patients, especially those treated with more injections. This assumption stemmed from the fact that inhibition of VEGF, which exhibits neuroprotective and anti-apoptotic properties, could contribute to the degeneration of the nerve fiber layer [18]. Intravitreal injections may also be associated with at least a short-term increase in intraocular pressure (IOP), a well-known risk factor for glaucoma, manifested by atrophy of the retinal nerve fibers [23]. Since defects in this layer may suggest the diagnosis of glaucoma, it seems important to detect them, as well as AMD, as it is characterized by a similarly increased prevalence in older age groups [22]. However, we did not focus on IOP in this study. Although we found a 0.17 µm decrease in RNFL thickness for each anti-VEGF injection in the wAMD group, this effect was not statistically significant (*p* = 0.109). We are aware that the potential changes in the RNFL in our treatment group could have been influenced not only by the number of injections received, but also by changes in the shape of individual layers resulting from the disease itself. Demir et al., who examined the inner layers of the retina in eyes with wAMD in the area of the macula unaffected by lesions, found no significant differences compared to the other, untreated eyes [28]. To minimize the potential artifacts related to incorrect contouring of individual retinal layers by the device software, each OCT scan was analyzed by the lead author (M.W.), with manual corrections applied when necessary (43.4% of all OCT scans). However, our observations indicate that software layer delineation errors mainly affect the outer layers of the retina, including Bruch’s membrane, retinal pigment epithelium, and photoreceptors, and occur mostly due to drusen or macular neovascular membrane irregularities.

It is well-established that anti-VEGF drugs in intravitreal therapy significantly reduce the thickness of the central retina. This is a desirable effect of treatment, associated with a reduction in the activity of the neovascular membrane in the macula. Since there were individuals in our study group who received a varying number of injections, it is important to note that the outcome may have been influenced by concomitant retinal edema resulting from an active neovascular membrane, especially among patients at the beginning of therapy who have received only the first injection (n = 10).

Although the differences in the RNFL layer did not turn out to be statistically significant, the differences in the GCL did (*p* < 0.001). Since the RNFL layer is made up of GCL (retinal ganglion cell) axons, the statistically significant effect of RNFL thickness on GCL found in our study may be not surprising (*p* = 0.002). The control group in our study was characterized by thicker ganglion cell layers compared to the treatment group. These findings seem to be similar to Beck et al., who also reported a lack of differences in RNFL and a thinner GCL in wAMD eyes treated with anti-VEGF compared to the fellow untreated eyes [29]. Moreover, Zucchatii et al. found slightly thicker GCL layers in non-advanced stages of dAMD compared to neovascular AMD. The authors explained GCL loss in more advances stages of AMD (both neovascular and atrophic) in two ways: as transneuronal degeneration due to input disturbances related to photoreceptors’ dysfunction, or simply due to hypoperfusion of the inner retinal layer as a result of macular senescence [13,26]. The first mechanisms may find confirmation in our study, as we also observed worse BCVA in the treatment group compared to the control group (0.58 and 0.84, respectively, *p* < 0.001), which may indirectly indicate damage of the retina outer layers, including photoreceptors, especially since BCVA turned out to be a factor with no statistically significant effect on the innermost layer of the retina, the RNFL (*p* = 0.341). The lack of BCVA influence on RNFL may also suggest that changes in photoreceptors do not affect fiber thickness, at least in the early stage of the disease presented by our patients, as their visual acuity had to allow them to maintain fixation during OCT. This finding may also confirm the thesis about the chronology of symptoms, originating within the outer layers of the retina.

In our study, sex also turned out to be statistically insignificant in terms of the thickness of the retinal nerve fiber layer in the macula. In our previous research, evaluating peripapillary retinal nerve fibers in AMD patients, we found significantly thinner RNFLs in men, which seemed consistent with a large study by Li et al. [19,30]. Although men in the current study had a macular RNFL 2.62 μm thinner than women, the difference was not statistically significant.

Among the analyzed factors, the only one that had a significant effect on the thickness of the RNFL layer was age. Patients over 73 years of age had a 3.35 µm thinner RNFL than younger patients (*p* = 0.042). The influence of age on the thinning of the inner layers of the retina has been documented by other researchers, which confirms the validity of including this factor in statistical analyses when conducting research on geriatric diseases [31,32].

Nonetheless, this study has a number of limitations. Firstly is the lack of long-term follow-up and the fact that the length of the treatment period is not considered in the statistical calculations. This is due to the lack of data on the time of injections received by patients in private centers before starting treatment in our clinic. Secondly, we did not consider the types of anti-VEGF drugs in the analysis, although different medications could potentially have varying effects on the analyzed retinal layers [23]. As we mentioned before, all three types of anti-VEGF agents have different biological activity, with bevacizumab and ranibizumab binding VEGF-A, and aflibercept binding VEGF-A, VEGF-B, PIGF-1, and PIGF-2, which could implicate different tissue responses to these compounds [2,10]. Our study group was relatively heterogeneous, with some patients receiving more than one drug: six patients (11.32%) were subjected to combined ranibizumab and aflibercept therapy, five patients (9.43%) were treated with bevacizumab and aflibercept, and one patient received bevacizumab, ranibizumab, and aflibercept (1.89%). Due to the small size of the study group, it was impossible to divide it into subgroups. However, although most of the studies on this subject included patients in monotherapy, there are also studies which include patients with combined anti-VEGF therapy (bevacizumab and ranibizumab or bevacizumab, ranibizumab, and aflibercept) [19,22,28].

In addition, we did not consider the physiological asymmetry between the eyes. However, we tried to minimize the impact of this factor by selecting patients with similar refractive errors, finding no statistically significant difference in the spherical equivalent of refractive errors between the study group and the control group (0.47 (SD 1.76) and 0.46 (SD 1.78) diopters, respectively, *p* = 0.978).

## 4. Materials and Methods

### 4.1. Patients

The study involved 106 eyes of 53 people over 45 years of age, treated in the national Drug Program for the Treatment of the Wet Form of Age-Related Macular Degeneration at the Department of Ophthalmology, Poznan University of Medical Sciences, between November 2020 and February 2022 (Figure 4). The inclusion criteria for the study were: the presence of unilateral wAMD treated with anti-VEGF injections according to the pro re nata regimen (bevacizumab, ranibizumab, or aflibercept, both monotherapy and combined—see Section 4.3.5), regardless of the number of injections received (patients at various stages of treatment—injections administered before treatment in our clinic were included in the total number of injections received), spherical equivalent of refractive error of +4.50 to −4.00 D, no systemic diseases other than controlled hypertension, and no history of cardiovascular events. The other eyes, with non-advanced dry AMD, served as a control group. In turn, advanced macular disease, preventing fixation on a given point during the optical coherence tomography examination, including atrophy in the case of dAMD, hemorrhage in the macula, or fibrosis in wAMD, excluded patients from participation in the study. In addition, exclusion criteria covered other potential macular diseases, all grades of diabetic retinopathy, a history of optic nerve disease including glaucoma, and its risk factors such as the presence of pseudo-exfoliation syndrome or intraocular pressure (IOP) higher than 21 mmHg, history of IOP-lowering medications use, and previous intraocular procedures other than cataract surgery performed at least 6 months before the examination. Each patient gave their written informed consent to participate in the study. The patients underwent a full ophthalmologic examination, including best-corrected visual acuity (BCVA) on Snellen charts, slit-lamp examination of the anterior eye segment, assessment of intraocular pressure, examination of the fundus after 1% tropicamide administration, and optical coherence tomography.

### 4.2. Optical Coherence Tomography

OCT tests were performed using a Topcon DRI OCT Triton device (TOPCON CORPORATION, Tokyo, Japan, Software: IMAGEnet6 for Triton Version 1.02.2 (1.34.2.19774) based on the Rescan 3D (H) protocol. They included scans of the macular area, 7 × 7 mm in dimensions, with automatic division into individual retinal layers. The layers analyzed were: the nerve fiber layer (RNFL), which covers the software-defined area from the inner limiting membrane (ILM) to the border between RNFL and GCL, and the retinal ganglion cell layer (GCL), which covers the area from the border between RNFL and GCL to the border between the inner plexiform layer (IPL) and the inner nuclear layer (INL), with the possibility of manual correction of the designated borders (which was used in 46 out of 106 scans). Scans with a minimum signal strength above 55 and without motion artefacts were included in the study (Figure 5).

### 4.3. Statistical Analysis

#### 4.3.1. List of Abbreviations of Statistical Measures 

*M*—mean

*SD*—the standard deviation

N—sample size

n—group size

*R^2^_conditiona_*_l_—conditional determination coefficient

*R^2^_marginal_*—marginal determination coefficient

σ^2^—mean squared error

τ_00 time—_inter-subject variance for time grouping factor

*N_id_*—population-level variable sample size

ICC*—*intraclass correlation coefficient

*p*—*p*-value

*CI*—95% confidence interval

ℤ^+^—space of positive integer values

ℝ^+^—space of positive real values

*t—*the *t*-test statistic

α—significance level

#### 4.3.2. Methodology

The significance level of the statistical tests in this analysis was assumed at α = 0.05.

#### 4.3.3. Regression Analysis

Examining the effects of multiple factors on the dependent variable is possible based on a regression model. However, the linear least squares regression model assumes that all observations are independent, including eyes of the same subject, which may imply correlation. A way to account for potential correlation was to estimate an additional subject-to-subject variance parameter. In other words, we were allowing for a random effect in which each subject contributes to the overall variability (Roback, 2021 [33]). In this case, the multilevel model was applied.

We fitted a linear mixed model, estimated using restricted maximum likelihood (REML) and the *nloptwrap* optimizer. The model included the patient ID as a random effect.

To perform a power analysis for a mixed model, a simr* package (version 1.0.6; Green P, MacLeod CJ, 2016 [34]) was used. The power was estimated at a range of sample sizes by the powerCurve() function. The power analysis based on the simulation showed that a sample size of 52 patients with repeated measurements was sufficient to achieve 80% power for multilevel regression models with two and three predictors (for the dependent variable RNFL, a spot sample size of N = 42 was sufficient to achieve a satisfactory level of power (80%)).

#### 4.3.4. Statistical Environment

Analyses were conducted using the R Statistical language (version 4.1.1; R Core Team, 2021 [35], Vienna, Austria) on Windows 10 × 64 (build 19044), using the lme4 (version 1.1.27.1; Bates D et al., 2015 [36]), Matrix (version 1.3.4; Bates D, Maechler M, 2021 [37]), ggeffects (version 1.1.1; Lüdecke D, 2018 [38]), sjPlot (version 2.8.10; Lüdecke D, 2021 [39]), report (version 0.5.1.3; Makowski D et al., 2021 [40]), ggstatsplot (version 0.9.3; Patil I, 2021 [41]), psych (version 2.1.6; Revelle W, 2021 [42]), and readxl (version 1.3.1; Wickham H, Bryan J, 2019 [43]) packages.

#### 4.3.5. Sample Characteristics 

The study sample included the results of eye examination of 53 subjects: 33 (62.3%) women and 20 (37.7%) men. One eye of each subject received treatment (injections of one or more drugs—bevacizumab, ranibizumab, or aflibercept), while the other eye remained untreated and was considered as a reference group. Twenty-eight (52.8%) left eyes and 25 (47.2%) right eyes were treated.

A total of 618 injections were performed: 9 subjects (17.0% of the sample) received a total of 54 injections (8.7%) of bevacizumab (*M* = 6.0, *SD* = 2.46 injections per person), 7 subjects (13.2% of the sample) received 50 (8.1%) injections of ranibizumab (*M* = 7.1, *SD* = 2.70 injections per person), and 50 subjects (94.3% of the treatment group) were treated with 514 (83.2%) injections of aflibercept (*M* = 10.28, *SD* = 7.50 injections per person).

For more information on the characteristics of the sample, see Table 5.

## 5. Conclusions

The morphology of the inner layers of the retina in dry and wet AMD seems to differ, possibly due to differences in the etiopathogenesis of these two forms of the disease. In our study, the retinal ganglion cell layer was thinner in the wAMD eyes compared to the fellow eyes (with non-advanced dry AMD), while the nerve fiber layer was not statistically different in both groups. The number of anti-VEGF injections had no effect on the thickness of the macular nerve fiber layer.

## Figures and Tables

**Figure 1 ijms-24-00402-f001:**
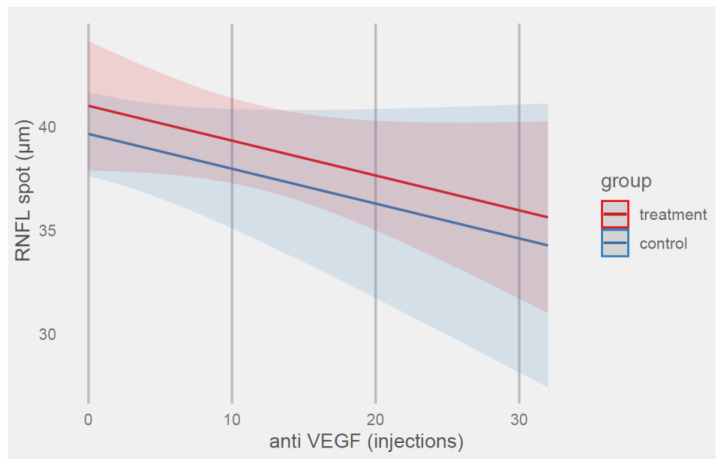
Prediction of RNFL thickness values (μm) by model 1 with anti-VEGF and group terms.

**Figure 2 ijms-24-00402-f002:**
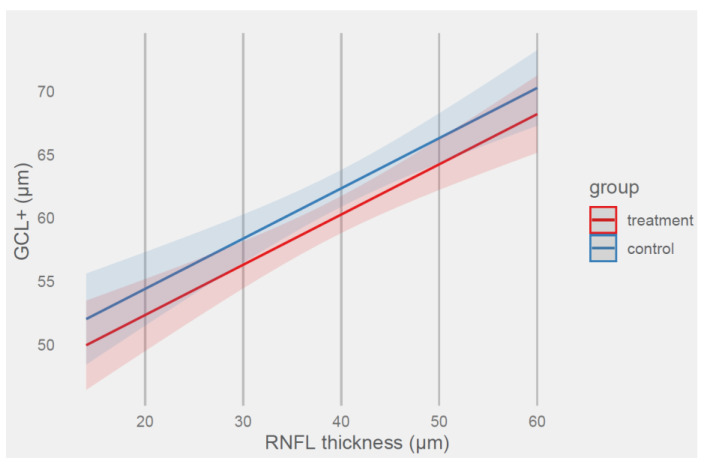
Prediction of retinal ganglion cell layer thickness (μm) by model 2 with RNFL thickness and group terms.

**Figure 3 ijms-24-00402-f003:**
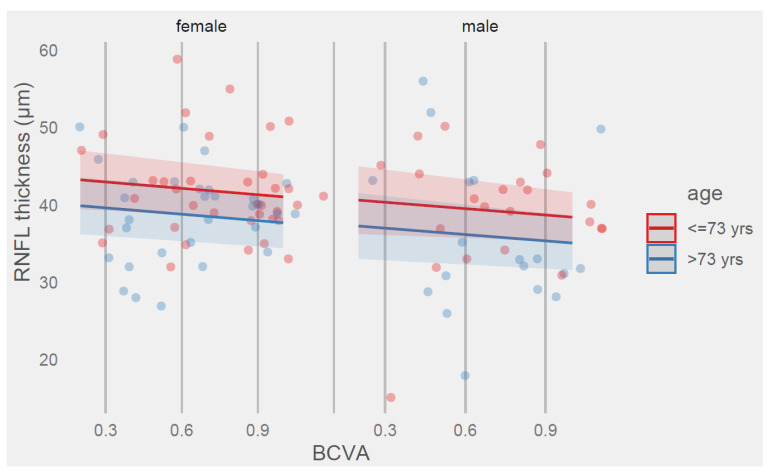
Prediction of RNFL thickness values (μm) by model 3 with BCVA, age, and sex terms.

**Figure 4 ijms-24-00402-f004:**
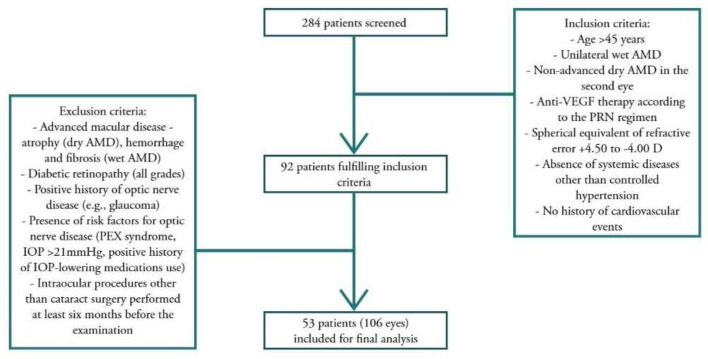
Study design.

**Figure 5 ijms-24-00402-f005:**
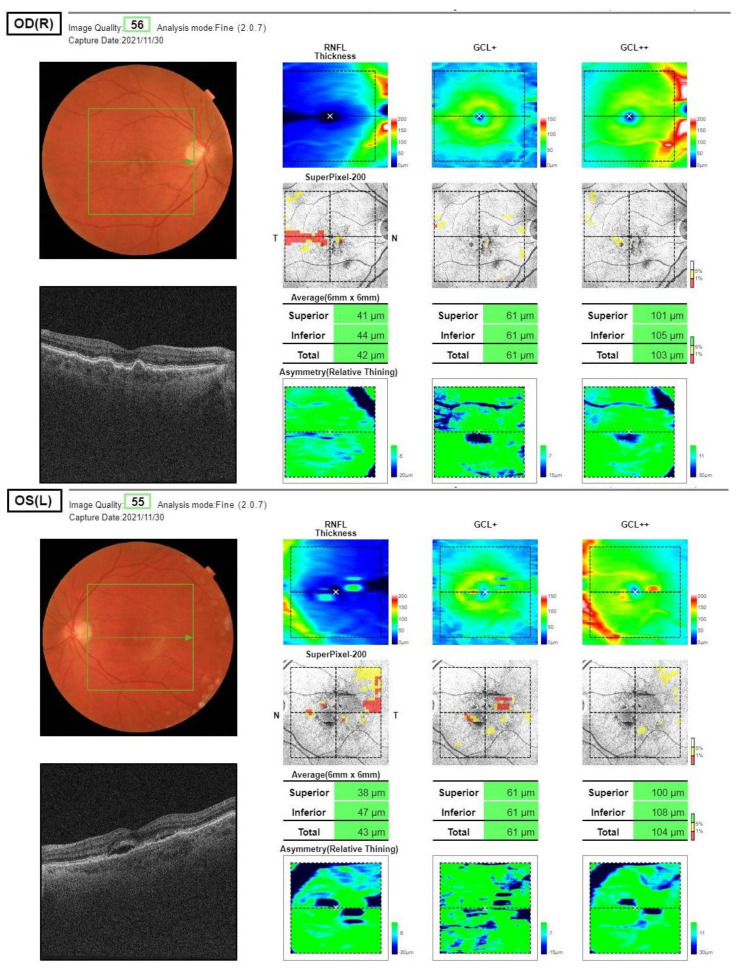
Optical coherence tomography, macula 3D scans (7.0 × 7.0 mm; 512 × 256), both eyes of the same patient. Analyzed values: RNFL total (covers the software-defined area from the inner limiting membrane (ILM) to the border between RNFL and GCL) and GCL+ total (covers the area from the border between RNFL and GCL to the border between the inner plexiform layer (IPL) and the inner nuclear layer (INL)).

**Table 1 ijms-24-00402-t001:** Results of fitting model 1.

Predictors	RNFL Thickness (μm)		
	Estimates	*CI*	*p*
(Intercept)	41.01	37.89–44.14	<0.001
Group (control)	−1.35	−4.60–1.89	0.409
Anti-VEGF	−0.17	−0.37–0.04	0.106
Random effects			
σ2	32.55		
τ_00 id_	22.18		
ICC	0.41		
*Nid*	53		

**Table 2 ijms-24-00402-t002:** Results of fitting model 2.

Predictors	GCL (μm)		
	Estimates	*CI*	*p*
(Intercept)	44.44	39.24–49.64	<0.001
Group (control)	2.06	0.75–3.38	<0.001
RNFL thickness	0.40	0.27–0.52	0.002
Random effects			
σ2	11.61		
τ_00 id_	16.61		
ICC	0.59		
*Nid*	53		

**Table 3 ijms-24-00402-t003:** Results of fitting model 3.

Predictors	RNFL Thickness (μm)		
	Estimates	*CI*	*p*
(Intercept)	43.81	39.05–48.58	<0.001
BCVA	−2.74	−8.44–2.95	0.341
Age (>73 years)	−3.35	−6.57–−0.13	0.042
Sex (male)	−2.62	−5.93–0.68	0.119
Random effects			
σ^2^	34.22		
τ_00 id_	17.09		
ICC	0.33		
*Nid*	53		

**Table 4 ijms-24-00402-t004:** Predictions for RNFL thickness based on BCVA lower and upper sample size range {0.2, 1.0} within age and sex variables.

BCVA	Age, Years	Sex	RNFL, μm	*95% CI*
0.20	≤73	Female	43.27	[39.47, 47.06]
1.00	≤73		41.07	[38.13, 44.01]
0.20	>73		39.93	[36.25, 43.59]
1.00	>73		37.72	[34.44, 41.00]
0.20	≤73	Male	40.64	[36.27, 45.02]
1.00	≤73		38.44	[35.26, 41.63]
0.20	>73		37.29	[33.05, 41.54]
1.00	>73		35.10	[31.61, 38.58]

**Table 5 ijms-24-00402-t005:** Characteristics of the continuous variables of the sample, N = 106, n = 53.

Parameter	Measure	Group	Value Distribution*, M* (*SD*) ^1^	*p*
Age	Years	-*	73.02 (7.42)	-
BCVA	-	Treatment Control	0.58 (0.2) 0.84 (0.2)	<0.001
S.E.	Dpt	Treatment Control	0.47 (1.76) 0.46 (1.78)	0.978
Anti-VEGF	Injections per subject	Treatment Control	11.66 (9.10) 0 (0)	<0.001
Time since diagnosis	Months	-*	21.13 (18.37)	-
GCL+	µm	Treatment Control	59.91 (6.94) 62.21 (5.44)	0.060
RNFL thickness	µm	Treatment Control	39.06 (8.69) 39.66 (5.84)	0.676

^1^ None of the variables on the continuous scale had pronounced values for skewness (>2.0) or kurtosis (>7.0), so a central measure of the tendency of the variables was given in the form of the mean along with the standard deviation; -* applies to both groups together (treatment and control).

## Data Availability

Not applicable.

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
