# Peer review of "Examination of Inner Retinal Layers in Unilateral Wet Age-Related Macular Degeneration Treated with Anti-VEGF, Compared to Fellow Untreated Eyes"

_ijms, 2022, doi:10.3390/ijms24010402_

Round 1

Reviewer 1 Report

Małgorzata Wichrowska and coworkers reported a characterization study differentiating RNFL(retinal nerve fiber layer) and GCL( ganglion cell layer) thickness within the macula region between the study groups wet AMD treated with antiVEGF vs Control. The authors further checked the influence of various factors such as BCVA, sex, age, the number of injections. Overall, the study is well-designed and well-conducted. The manuscript presentation and datasets are clear and well-presented. This work will have an impact on real-world clinical settings and well deserved to be published in the IJMS journal. This paper adds to the existing knowledge in the field.

However, I have the following comments, which may be addressed before acceptance in the journal.

Introduction:

1.     Authors introduced the topic well however they need to explain the rationale for modeling and why specific those models were used and in the previous literature, such models exist.

2.     A consent statement may be conveniently moved to the end of the paper. Not necessary in the introduction.  

3.     Line 41-51 the recent references may be useful.

https://www.mdpi.com/1999-4923/14/10/2133

https://www.sciencedirect.com/science/article/pii/S135964462200160X

https://www.nature.com/articles/s41572-021-00265-2

https://www.sciencedirect.com/science/article/pii/S1572100020303847

4.     Please make it clear if single anti-VEGF or combination studies are available previously on this subject.

Results

1.     Check if your equations can be numbered and cited in the text for readability.

Methods Patients

1.     I recommend adding a flow chart/graphic for better clarity between groups including inclusion and exclusion criteria in the same.

2.     It is unclear which anti-VEGF you are referring to. Are you referring to all 3 (bevacizumab, ranibizumab, or aflibercept)? combined? individual

Results

1. OCT images are beneficial for the reader if you can show the difference, at least representative.

Discussion:

“Second, we did not consider the types of anti-VEGF drugs in the analysis, although different medications could potentially have varying effects on the analyzed retinal layers”

1.     Please elaborate on the differences between molecules used in the manuscript. As you already know the variations. https://www.mdpi.com/1422-0067/23/16/9424

If you failed to find previous studies, please make this clear.

 Reference formatting changes are required, and spell check is required. Provide a list of acronyms at the end. 

Author Response

Response to Reviewer 1:

I would like to thank you for your review and positive attitude to our work. I appreciate your thorough evaluation of our paper and interesting comments. Our answers to your points are as follows:

  1. Introduction:

1.1.    Authors introduced the topic well however they need to explain the rationale for modeling and why specific those models were used and in the previous literature, such models exist.

Motivation, in terms of content, is given in 4.3.3 of the article. A model with such a design allows for the calculation of the effects of many variables, taking into account repeated measurements in the form of a random effect for individual people. These models are described in the example sources:

https://bookdown.org/roback/bookdown-BeyondMLR/ (Chapter 8),

1.2.     A consent statement may be conveniently moved to the end of the paper. Not necessary in the introduction.  

As you kindly suggested, I have moved the consent statement to the end of the paper (lines 467-470).

1.3.     Line 41-51 the recent references may be useful.

https://www.mdpi.com/1999-4923/14/10/2133

https://www.sciencedirect.com/science/article/pii/S135964462200160X

https://www.nature.com/articles/s41572-021-00265-2

https://www.sciencedirect.com/science/article/pii/S1572100020303847

Thanks for the advice. I carefully read all these suggested works and used some of them to improve our introduction (lines 48-57, 71-78). Moreover, I have made the necessary corrections regarding the numbering of references.

1.4.     Please make it clear if single anti-VEGF or combination studies are available previously on this subject.

According to our knowledge, most studies were conducted using anti-VEGF monotherapy. However, we also found studies based on combined therapy. As you suggest, we added this information (lines 323-326).

  1. Results: Check if your equations can be numbered and cited in the text for readability.

The numbers for individual composite models (e.g. mod.2) are given, and solely referred to in the content of the analysis. Since the level 1 and 2 equations are components of the composite model, there is no need to number them.

  1. Methods Patients
  2. 1.     I recommend adding a flow chart/graphic for better clarity between groups including inclusion and exclusion criteria in the same.

I agree, it organizes the information well for the reader. We have added the graphic as a figure 4.

3.2.     It is unclear which anti-VEGF you are referring to. Are you referring to all 3 (bevacizumab, ranibizumab, or aflibercept)? combined? Individual

I agree that it has not been clearly shown here that we are dealing with a heterogeneous group of patients treated both in monotherapy and in combination therapy with two or even all three agents. Thus, I have added this information in lines 338-340. It is also described in more detail in section 4.3.5 (Sample characteristics), and discussion (lines 319-322).

  1. Results: OCT images are beneficial for the reader if you can show the difference, at least representative.

I have some doubts regarding adding OCT images to show the differences, as differences could only be shown as a result of statistical analysis. However, the addition of OCT reports can indeed familiarize readers with the protocol used, as well as indicate the values that we used for statistical analysis. Hence, the reports were added as fig. 5. (lines 375- 381).

  1. Discussion: “Second, we did not consider the types of anti-VEGF drugs in the analysis, although different medications could potentially have varying effects on the analysed retinal layers”. Please elaborate on the differences between molecules used in the manuscript. As you already know the variations. https://www.mdpi.com/1422-0067/23/16/9424 

We are very grateful for getting acquainted with our work so far. We added an extra paragraph containing this information in the introduction and emphasized it in the discussion (lines 69-78, 315-319). However, this was not intended to be an extensive review, and we believe that detailed information on this subject can be easily found in other sources, including those cited in the literature. In addition, we will develop this topic further in our next review article, which will close my doctoral dissertation.

  1. Reference formatting changes are required, and spell check is required. Provide a list of acronyms at the end. 

All references are cited according to Vancouver style and cited using PubMed citation. We have renumbered references in text and on references list due to the introduction of new suggested papers. A spell check has been performed by a qualified English language editor, with the list of acronyms has been provided at the end of the article.

Reviewer 2 Report

Wichrowska et al. analysed inner retinal layers in unilateral wet age-related macular degeneration (wAMD) treated with anti-VEGF compared to untreated eyes of the same patients. They found thinner ganglion cell layers GCL in wAMD vs. dAMD (p<0.001). Retinal nerve fiber layers were not significantly different. Age >73 years was the only factor significantly affecting the thickness of the retinal nerve fiber layer. 

The manuscript is overall well written. Some minor modifications are required:

Line 57: include the Refs in the sentence.

Table 2 should be separated from the main text instead of the surrounding.

Line 151/Line 163: Cause and consequence are inverted? Mathematical it seems to be correct but biological it would be the other way round.

Line 188: spelling error

Line 258: please cite authors consistently as Last name et al., (Beck et al., instead of Beck M et al) throughout the ms.

Author Response

I would like to thank you for your review and positive attitude to our work. I appreciate your thorough evaluation of our paper, our answers to your points are as follows:

Line 57: include the Refs in the sentence.

Thank you for your valuable point, I have already corrected the error.

Table 2 should be separated from the main text instead of the surrounding.

Yes, indeed, I have already corrected the error.

Line 151/Line 163: Cause and consequence are inverted? Mathematical it seems to be correct but biological it would be the other way round.

In fact, it does not seem logical from a medical point of view, but this statistical model is directed, and it cannot be said inversely that changes in the GCL affect the RNFL. I will remove that sentence to avoid confusing the readers, since they can find all this data in the table anyway.

Line 188: spelling error

Thank you for your valuable point, we have already corrected the error.

Line 258: please cite authors consistently as Last name et al., (Beck et al., instead of Beck M et al) throughout the ms

We have corrected the citation as per your recommendation.

Reviewer 3 Report

I’ve read with attention the paper of Wichrowska et al. that is potentially of interest. The background and aim of the study have been clearly defined. The methodology applied is overall correct, the results are reliable and adequately discussed. I’ve only some minor comments:

- It is not clear if the authors have calculated the patient sample size or the study power

- Given the focus of the journal on molecular sciences, the reader could expect more biochemical speculation than clinical data per se. In this context, the authors should enrich introduction and discussion on the molecular basis of the studied diseases and the tissue changes induced by the treatment.

Round 2

Reviewer 1 Report

The authors carefully addressed all comments. Acceptable in current form.